# Reconfigurable photoactuator through synergistic use of photochemical and photothermal effects

Markus Lahikainen [1], Hao Zeng [1] & Arri Priimagi[1]

A reconfigurable actuator is a stimuli-responsive structure that can be programmed to adapt different shapes under identical stimulus. Reconfigurable actuators that function without control circuitry and are fueled remotely are in great demand to devise adaptive soft robotic devices. Yet, obtaining fast and reliable reconfiguration remains a grand challenge. Here we report a facile fabrication pathway towards reconfigurability, through synergistic use of photochemical and photothermal responses in light-active liquid crystal polymer networks. We utilize azobenzene photoisomerization to locally control the *cis*-isomer content and to program the actuator response, while subsequent photothermal stimulus actuates the structure, leading to shape morphing. We demonstrate six different shapes reconfigured from one single actuator under identical illumination conditions, and a light-fueled smart gripper that can be commanded to either grip and release or grip and hold an object after ceasing the illumination. We anticipate this work to enable all-optical control over actuator performance, paving way towards reprogrammable soft micro-robotics.

---

[1] Laboratory of Chemistry and Bioengineering, Tampere University of Technology, P.O. Box 541, FI-33101 Tampere, Finland. Correspondence and requests for materials should be addressed to H.Z. (email: hao.zeng@tut.fi) or to A.P. (email: arri.priimagi@tut.fi)

For the past decades, great efforts have been put into investigating stimuli-responsive soft materials and their use in designing soft actuators capable of complex, rapid, and reversible macroscopic movements[1,2]. The inspiration for the soft actuator research often times comes from natural species that are capable of adaptive shape changes, diverse geometric morphing, and highly efficient locomotion[3,4]. Stimuli-responsive soft actuators have proven their potential in realizing bio-inspired soft robots that are miniature in size and can be fueled remotely and wirelessly to yield complex movements[5–7]. Compared to synthetic, human-made devices, the sophistication of natural species is overwhelming: even the most primitive species can have multiple degrees of freedom in their movement, and reconfigurable response under the same environmental stimulus. Most artificial actuators, on the other hand, display only one specific kind of deformation under a specific stimulus (e.g., light, magnetic field, electric voltage)[8–10], and obtaining reconfigurable actuation under constant stimulus is a formidable challenge. Achieving this goal would pave way towards adaptive, reprogrammable soft micro-robotics.

Among different stimuli, light has arisen as a particularly attractive alternative, due to its high degree of spatial and temporal control over properties such as wavelength, intensity, and polarization. Several photoresponsive molecules[11,12] and material systems[13–15] such as hydrogels[7,16], carbon-nanotube-based actuators[17], and shape memory polymers[18] have been explored for devising soft actuators, yet photoresponsive liquid crystal elastomers and polymer networks (referred to as LCNs from hereon) are in many cases particularly intriguing[19]. LCNs combine the anisotropy of liquid crystals and the elasticity of loosely crosslinked polymeric materials, and by "programming" the molecular alignment pattern during the fabrication process (photopolymerization), an overwhelming variety of light-induced shape changes (e.g., contraction, bending, coiling, buckling, etc.) can be obtained[20–23]. In contrast to hybrid actuators relying on, e.g., bilayer structures to obtain efficient actuation[7,17], LCNs allow complex three-dimensional deformation in a monolithic film with uniformly distributed chemical composition[19].

The term reconfigurable actuation is often times used to refer to the ability to reversibly switch between different shapes upon external stimulus[14,15]. However, conventional stimuli-driven actuators, LCNs among other materials, even if reversible, typically exhibit only one pre-determined state of deformation under one specific stimulus. Hence, to obtain multiple shape changes, the stimulus has to be tuned, which in case of light-triggered actuation can refer to, e.g., changing the light intensity or its spatial/temporal distribution. Obtaining multiple shape changes upon an unchanged stimulus thus requires a programming step during fabrication (such as controlling the photo-polymerization process[24]) which eventually results in different samples with identical composition but distinct actuation modes. In contrast, reconfigurable actuators, as we define them, have the ability to obtain multiple shape changes from one single sample upon one identical stimulus. Previous reconfiguration strategies have been based on dynamic covalent bonds[25–29], acidic patterning[30], or base treatment[31]. All these methods rely on chemical processing to modify material properties like alignment, elastic modulus, and absorption, while the diversity of reconfigured shape change is limited. The lack of facile reconfiguration schemes is a significant drawback from the perspective of light-powered micro-robotics[32] which would benefit from fast and reliable control over actuation modes. A light-fueled soft actuator that would be reconfigurable, i.e., exhibit more than one type of shape change under the same light stimulus (same intensity, wavelength, polarization, etc.), and at the same time being fast and reliable, opens new horizons to realize flexible micro-devices with well adjustable performance.

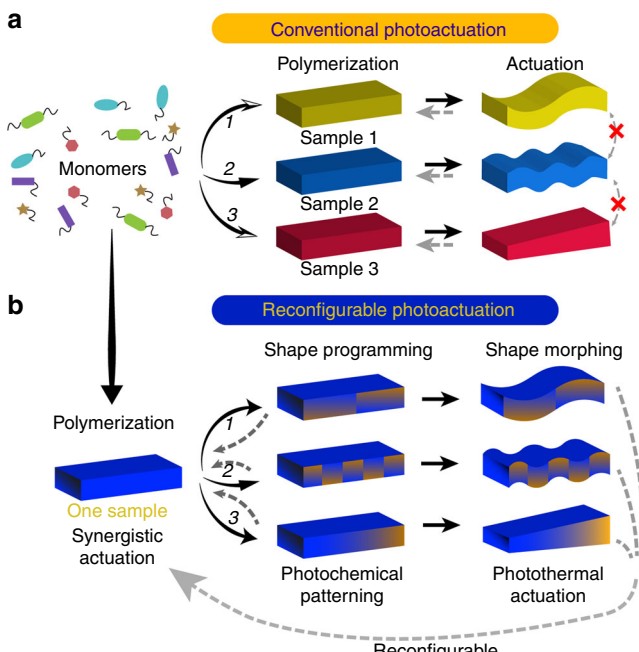

**Fig. 1** Actuator concept. **a** Conventional stimuli-responsive actuators require a programming process in fabrication which eventually results in different samples with distinct actuation modes. **b** A reconfigurable photoactuator capable of multiple shape changes from one single sample upon one identical stimulus is achieved using photochemical trigger for shape programming and photothermal effect for shape morphing

Herein we present a fabrication strategy for solving this challenge through a facile pathway towards reconfigurable photoactuation. Our concept is based on synergistic use of the photochemical and photothermal actuation schemes, a combination that we show to yield "actuation on command". The photochemical and photothermal effects play distinct roles in the synergistic photoactuation in a liquid crystalline polymer network, the former being used for shape programming and the latter for shape morphing. We demonstrate six different reversible shapes taken by the same actuator within a few seconds of irradiation under identical illumination conditions. The concept is further utilized to design a smart gripping device that can be commanded to either hold or release the object after ceasing the irradiation. These results offer an approach to reconfigurable micro-robotics with reversible, fully optical control over the shape programming and shape morphing.

## Results

**The design concept of synergistic photoactuator.** Figure 1 shows the conceptual differences between conventional actuators and the reconfigurable actuation strategy proposed for the LCN actuator based on synergistic use of photochemical and photothermal effects. The light-induced deformations in LCNs can take place via two distinct mechanisms, being driven by either photochemical[33] or photothermal stimulus[20]. Photochemical LCN actuators rely on photoisomerization of azobenzene molecules[34], typically incorporated as crosslinks into the polymer network. The lifetime of the deformed shape is connected to the *cis*-isomer lifetime of the azobenzene used[35], and the actuator can be driven between two distinct states by irradiating it with two different wavelengths (see the schematic in Fig. 2a). Such systems allow for devising optically tunable photonic devices[36], and also enable efficient photoactuation in aqueous environment[37]. For soft robotic applications, the most efficient photoactuation scheme is

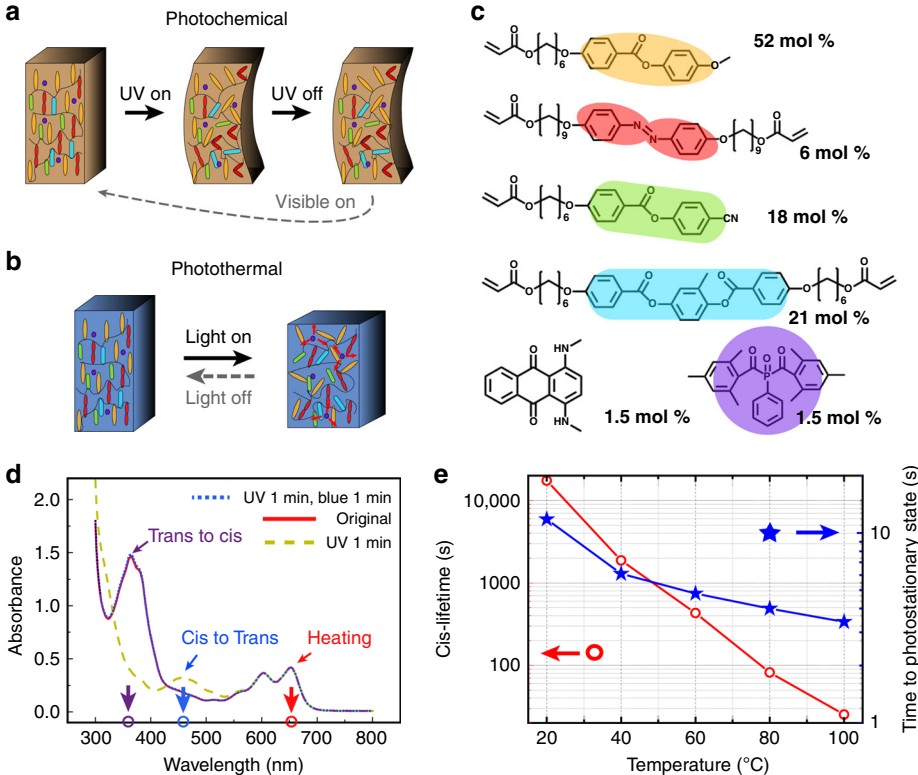

**Fig. 2** Design of a synergistic photoactuator. Schematic drawing of **a** photochemical and **b** photothermal actuation. **c** Chemical composition of the liquid crystal monomer mixture. **d**, UV–Vis spectra of LCN film containing both DB14 and azobenzene crosslinks. The arrows indicate the excitation wavelengths used in the actuation: 365 nm for *trans–cis* isomerization (purple arrow), 460 nm for *cis–trans* isomerization (blue arrow), and 660 nm for photothermal heating (red arrow). **e** *Cis*-lifetime during the thermal relaxation in the dark (*cis–trans*) and time to reach photostationary state upon UV irradiation (*trans–cis*) at different temperatures

based on converting light energy into heat, which allows for fast and dynamic (i.e., the system rapidly relaxes back to the initial state once the irradiation is ceased, see Fig. 2b) motions and cyclically actuated, small-scale photomobile devices[38,39]. The different actuation dynamics of photochemically and photothermally driven strip-like LCN actuators is highlighted in Supplementary Figure 1. Both actuation mechanisms have yielded a wealth of fascinating demonstrations on light-fueled movements[40–42]. In the following, we demonstrate the synergistic use of these two mechanisms in order to design photoresponsive materials with characteristics not met in purely photochemical or photothermal actuators.

The materials used and their composition within the LCN actuator are shown in Fig. 2c (see Methods for fabrication details). The photoactive units were selected based on the following criteria. First, the wavelengths with which to trigger *trans–cis* isomerization, *cis–trans* isomerization, and the photothermal effect must be spectrally separated (see Fig. 2d and Supplementary Figure 2). Second, the lifetime of the *cis*-isomer of the azobenzene crosslinks at the operating temperature (accounting for the temperature rise due to the photothermal effect) must be long enough, such that thermal back isomerization within the experimental time frame would be minimized. The ultraviolet–visible (UV–Vis) spectra of the polymerized LCN actuator are shown in Fig. 2d, confirming the spectral separation and efficient, reversible isomerization of the azobenzene crosslinks. Figure 2e shows that the *cis*-lifetime (see Supplementary Figure 3 for further details) of the azobenzene used scales exponentially with temperature. For example, at 40 °C the *cis*-lifetime exceeds 30 min, which is long enough to be considered as bi-stable within the time frame of photothermal actuation, as will

be detailed later on. The fraction of *cis*-isomer in the photostationary state is ca. 80% even at elevated temperatures (see Methods for further details). However, the time to reach the photostationary state decreases significantly with increasing temperature (Fig. 2e). Therefore, certain degree of temperature increase due to photothermal effect serves to accelerate the photochemical actuation.

In our synergistic photoactuation scheme (Fig. 1b), the *trans–cis* isomerization of azobenzene moieties is used to locally control the mechanical properties of the actuator (yielding no shape changes), while photothermal effect triggers shape changes under red-light illumination by releasing the stress generated by inhomogeneous *cis*-azobenzene distribution. As the reverse *cis–trans* isomerization can be induced by blue-green irradiation, the system can be brought back to its initial state upon irradiation within these wavelengths, and subsequently reconfigured with UV light to adapt any other shape under red-light illumination.

**Enhanced photoinduced bending in synergistic photoactuators**. We studied the photoinduced bending of the synergistic photoactuators in splay-aligned films with $4 \times 1 \times 0.02$ mm³ dimensions. In splay films, both photochemical and photothermal effects give rise to bending towards the same direction[43]—the planar-oriented side of the film—and can be easily characterized by measuring the bending angle as schematized in the inset of Fig. 3a. Upon photothermal heating (660 nm), the strip exhibits a gradual bending with magnitude increasing as a function of irradiation intensity (red circles in Fig. 3a), accompanied by a linear increase in temperature (blue dots in Fig. 3a). The speed of temperature increase/decrease upon light on/off does not depend

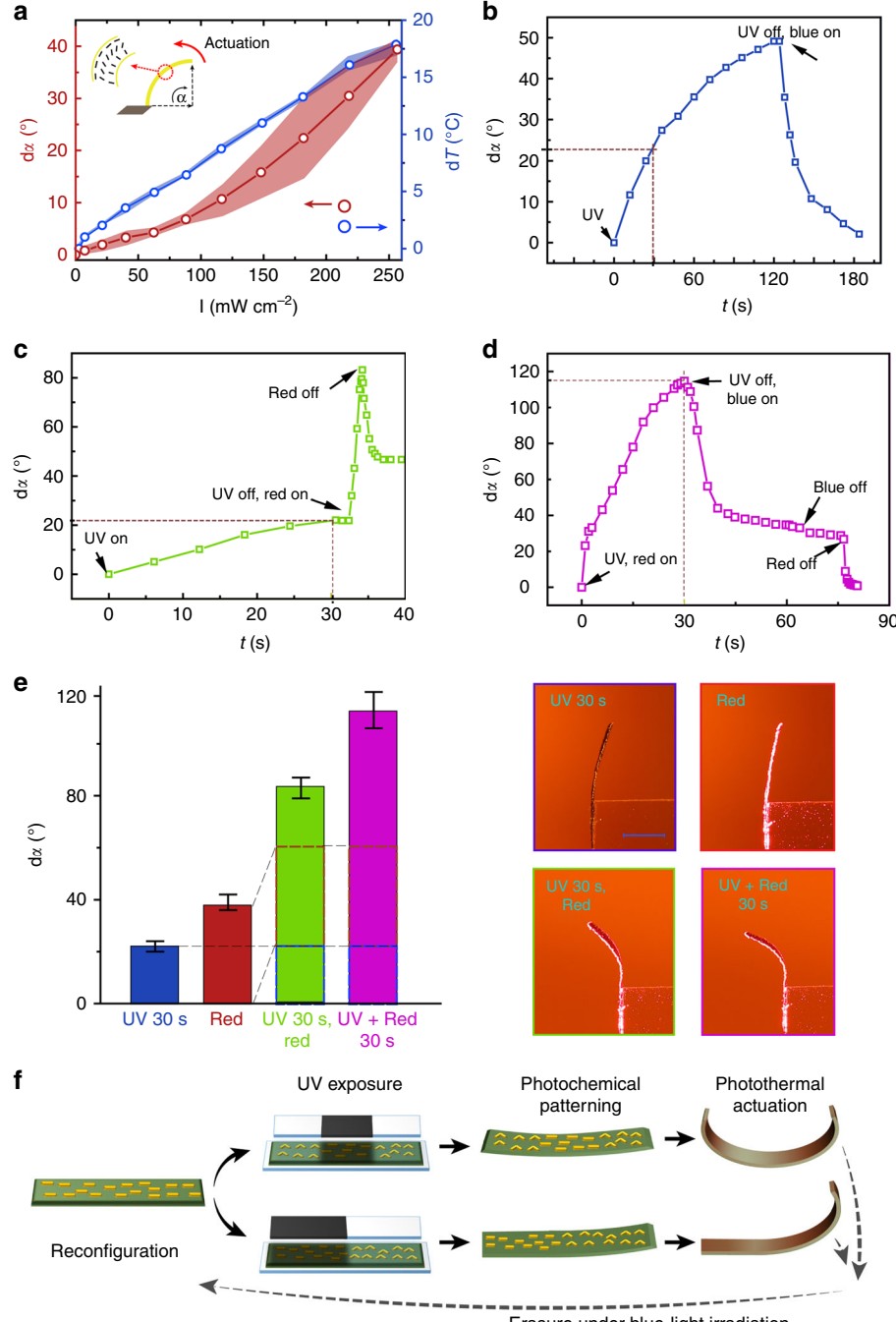

**Fig. 3** Photoinduced bending upon photochemical and/or photothermal triggering. **a** Change of bending angle d$\alpha$ and maximum enhanced temperature as a function of red-light irradiation intensity. The error bars indicate standard deviation for $n = 3$ measurements. Inset shows the schematic drawing of the measured bending angle and actuation direction in respect to the molecular alignment. **b** Photochemically induced bending upon subsequent UV and blue light excitations. **c** Synergistic actuation using subsequent UV and red-light illumination. **d** Synergistic actuation under simultaneous UV and red-light illumination. **e** Bending angle change upon different stimuli. Error bars indicate standard deviation for $n = 3$ measurements. Photographs show the deformation of the same sample upon different illumination. Scale bar: 2 mm. **f** Schematic drawing of reconfiguration strategy by synergistically using photochemical and photothermal responses

on irradiation intensity (Supplementary Figure. 4), and both bending and unbending take place relatively fast, within less than 2 s once starting/ceasing the irradiation. As shown in Supplementary Fig. 5a–c, the response time is similar for samples with different thicknesses. In the experiments that follow, we adopt a moderate intensity of 260 mW cm$^{-2}$, corresponding to 18 °C temperature increase at the equilibrium stage and ca. 40° of photothermally induced bending.

For photochemical actuation, 365 nm (9 mW cm$^{-2}$) and 460 nm (18 mW cm$^{-2}$) were used for inducing *trans–cis* and *cis–trans* isomerization, respectively. The reason for using such low intensity is that we want to fully decouple the photochemical and photothermal actuation processes and as can be seen from Supplementary Fig. 6, heating induced by UV or blue irradiation is negligible, less than 1 °C. The photochemically induced bending is markedly slow compared to the photothermal actuation: within

2 min of UV exposure, the strip slowly bends to 50°, and 60 s irradiation with blue light retains the original shape (Fig. 3b). The slow bending speed is consistent with literature on photochemically driven actuators[8,21,35].

We next combined the photochemical and photothermal effects, first subsequently and then simultaneously. In Fig. 3c, 30 s UV irradiation, leading to bending of 22°, is followed by the photothermal excitation. Such subsequent UV → red illumination yields rapid bending up to 83°, i.e., much larger than the bending induced by red light only (40°), without prior UV exposure (Fig. 3a). We attribute this to photochemically induced stress inside the polymer network which can be released by photothermal heating. Interestingly, when the red light is ceased, the strip relaxes only partially and stabilizes to 47° angle (cf. 22° bending angle prior to red-light illumination). This indicates that the photochemically induced deformation can be preserved and even amplified by the photothermal actuation. Finally, when illuminating the film simultaneously with UV and red light to activate both photochemical and photothermal mechanisms, the strip bends to 115° angle within 30 s (Fig. 3d), overwhelming the purely photochemical actuation (Fig. 3b) in both speed and magnitude.

The above-described experiments are summarized in Fig. 3e. Most importantly, combination of photochemical and photothermal actuation leads to a drastic increase in the bending angle—fivefold compared to the photochemically induced actuation and threefold compared to the photothermal one. Such enhanced photoinduced deformations are observed in samples with different thickness, as shown in Supplementary Figure 5d, which also illustrates that the bending angle increases with sample thickness. As a comparison, in carbon-nanotube-based photothermal actuators, both the degree of deformation and wavelength sensitivity decrease when increasing the thickness of the actuator[44]. Herein, we attribute the slightly enhanced actuation in thicker LCN strips (Supplementary Figure 5d) to changes in material rigidity upon changing the thickness. This is because the polymerization process itself depends on the sample thickness: the thicker the sample, the stronger the absorption due to the azobenzene units at the polymerization wavelength (420 nm; see Fig. 2d for the azobenzene absorption). Therefore, in thicker samples polymerization becomes less efficient, yielding softer samples with enhanced photoinduced bending. Figure 3e also reveals another interestingly fact: the enhancement arising from the photoisomerization and the photothermal heating can be achieved in different time domains. Therefore, photoisomerization-induced changes can be recorded/patterned into the polymer without introducing significant actuation, after which photothermal actuation can be applied to deform the structure without the need for further changes in isomerization (green column in Fig. 3e). The deformation is fully erasable by irradiation with blue light that induces cis-to-trans back isomerization. This mechanism immediately points towards an application in realizing light-driven reconfigurable actuators, based on the principle illustrated in Fig. 3f. Firstly, fully reversible photochemical pattering via mask exposure allows for local control over the cis-isomer content across the sample area, thus programming the actuator performance. Subsequently, photothermal heating stimulates the UV-encoded actuation capacity, leading to diverse shape morphing under identical light stimulus.

**Reconfigurable shape changes and programmable microrobotics**. To demonstrate reconfigurability based on synergistic photoactuation, we fabricated a planar-aligned LCN strip with $25 \times 1 \times 0.05$ mm$^3$ dimensions and reshaped it into six different geometries under identical, spatially uniform red-light illumination (Fig. 4). This is achieved by spatially patterning areas with

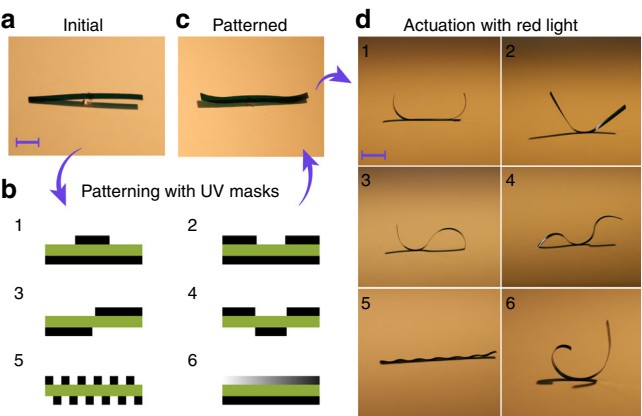

**Fig. 4** Reconfigurable shape morphing. **a** Optical image of the actuator at the initial stage. **b** Schematic of mask patterns. **c** Optical image of the actuator after UV exposure using mask no.1. **d** Photothermally induced shape morphing upon red-light illumination after different mask exposure. Scale bars: 5 mm

high cis-isomer content by masked UV exposure from either one side or both sides of the sample (Fig. 4b). As shown in Fig. 4c, the UV preirradiation gives rise to negligible shape change of the strip, even if the cis-content within the actuator is significantly altered (i.e., the photochemically induced stress is "hidden" inside the LCN actuator). Upon red-light illumination, the strip quickly deforms into different geometries determined by the UV preirradiation (Fig. 4d, Supplementary Movie 1). We note that all the shapes are obtained within the same strip after retaining the initial flat state by exposure with blue light to convert all the cis-azobenzenes back to the trans-form (see in Methods).

In 2017, we reported on a light-fueled gripping device that can autonomously recognize and manipulate objects based on their reflection/scattering properties[45]. The concept proposed here opens up a strategy to design smart grippers with reconfigurable action. This is demonstrated by the device shown in Fig. 5. The gripper is fabricated by fixing two splay-aligned LCN actuators on a tip of a capillary, with photothermal excitation (red-light irradiation) coming from the back along the capillary direction (see Supplementary Fig. 7). The gripper is designed such that it is open in the dark (Fig. 5a), closes under red-light illumination (Fig. 5b), and retains the original shape when ceasing the irradiation. By such photothermally activated gripping action, the device is able to clamp and hold a 60 mg object, 100 times heavier than the actuator itself. If the device is preirradiated with UV, the gripper deformation is much more pronounced under identical red-light illumination, as shown in Fig. 5c and as expected based on Fig. 3c. In this case, the gripper remains closed even after ceasing the red-light illumination, due to the release of the photochemically induced stress (Fig. 5d). Therefore, synergistic photoactuation allows us to program the gripping device to adapt an operation mode in which the object is gripped, lifted up, and dropped immediately when the energy source is removed (Fig. 5e, also see Supplementary Movie 2), or gripped, lifted up, and held in place even after ceasing the irradiation (Fig. 5f, Supplementary Movie 3). Most importantly, the device can be reconfigured with irradiation with blue light and commanded whether to grip and release or grip and hold, which is a unique ability in LCN photoactuators.

**Discussion**
The demonstrated reconfigurable photoactuator is unique compared to other dual-responsive liquid crystalline polymer systems reported in the literature[46–48], and to wavelength-selective

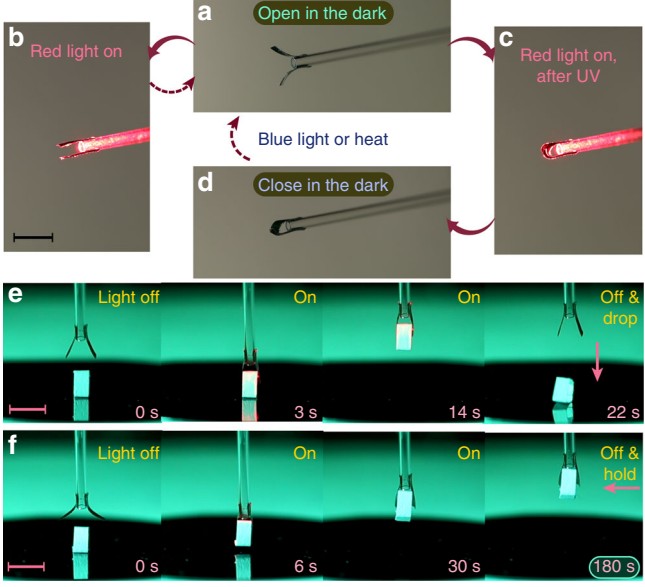

**Fig. 5** A reconfigurable micro-gripper. A gripper in its initial, open stage (**a**) closes upon red-light illumination (**b**) and returns back to the original state when turning off the light. After UV exposure, the gripper closes upon red-light illumination (**c**) and maintains the closed state even after ceasing the irradiation (**d**). **e** Series of optical images of the gripper operated in a grip-and-release mode, and **f** the same device reprogrammed to operate in a grip-and-hold mode. Object being lifted up in (**e**, **f**) is 12 mg in weight. All scale bars: 5 mm

carbon-nanotube-based photothermal actuators[17,44]. To the best of our knowledge, our strategy is only one where the photochemical and photothermal stimuli act in concert in order to enhance the overall performance of the system and bring about new, unprecedented capacities. We use photochemistry for reconfigurability and photothermal effect for shape morphing, i.e., both serve their distinct, specific purposes. Even if playing distinct parts, they are inherently integrated, as shown by the actuation speed and deformation amplitude experiments of Fig. 3.

In many cases, quantitative analysis of purely photochemically driven actuation is complicated, due to the fact that it is often accompanied by a certain degree of photoinduced heat generation. We would like to draw attention to the fact that careful characterization and full decoupling of the effects arising from photoisomerization and photothermal heating is necessary for understanding the overall dynamics of photoactuation. In our case, photothermal heating of 18 °C (shown in Fig. 3a) brings about dramatic changes in the photochemical actuation in the LCN material with glass transition temperature at around room temperature (see differential scanning calorimetry measurement in Supplementary Fig. 8). It enhances the actuation speed by a factor of 12 (bending up to 50° takes 2 min upon UV irradiation but only 10 s upon simultaneous UV–red irradiation, see Fig. 3b, d), and the deformation amplitude by a factor of 5 (Fig. 3e). We attribute this to photothermally induced softening of the polymer matrix which allows the bent *cis*-isomers to transfer the deformation from molecular level into deformation of the whole polymer network with higher efficiency. Other researchers have shown that cyclic *trans–cis–trans* isomerization is able to soften the polymeric matrix[49] and enhance the topological actuation in photoresponsive cholesteric coatings[50]. Cyclic isomerization can also bring about significant mass migration of polymer chains, allowing for photoinduced surface patterning on amorphous polymer thin films[51]. We anticipate that the synergistic use of photochemical and photothermal effects and the results

explicated in Fig. 3 can contribute to elaborating the photo-actuation process for many azobenzene-containing polymer networks and elastomers.

The reconfigurability is based on the spatial control over the concentration of *cis*-isomers, thus the stability of the reconfigured state is inherently connected to the *cis*-lifetime of the azobenzene crosslinks. Therefore, the stability of the deformed structure depends on the environmental conditions, being longer in normal room lighting than outdoors under direct sunlight or at elevated temperatures (Supplementary Fig. 9). With the present material composition, the bent film recovers 50% of its original shape within 9 min under laboratory lighting (Supplementary Fig. 9a), revealing the time limit for preserving the reconfigured state. This weakens the sustainability of the reconfigurable robotics, and for instance the gripper shown in Fig. 5f can hold the object only for ca. 5 min after ceasing irradiation. This limitation can be overcome using azobenzenes with long *cis*-lifetime, such as the *ortho*-fluorinated ones demonstrated by Hecht and coworkers[52,53]. Improving the temporal and thermal stability of the reconfigured state is an important next step in our research.

We also emphasize that apart from LCNs, various types of materials exhibit light-induced shape changes, such as hydrogels[7,16], temperature-responsive and shape memory polymers[18,54] functional carbon-nanotube-based bilayers[17], photochromic crystals[55], etc. While chemists and material scientists continuously extend the diversity of materials and shape changes, engineers and micro-roboticists are dedicated to issues such as energy consumption[56] and integration of the photomechanical units as part of active circuits[2]. We note that the physical principles underlying light-driven device/robot realization using different materials are rather similar, as exemplified by, e.g., self-shadowing-induced oscillation[17,57], snapping motion[58,59], and rolling locomotion[17,60], all achieved in both LCNs and bilayer actuators. Keeping this in mind, we believe that the reconfigurability concept proposed herein can also be extended beyond LCNs, and synergies can be sought for by combining different types of light-responsive materials to simultaneously harness the strengths of each.

We have demonstrated a method to achieve reconfigurable actuation by combining photochemical and photothermal responses within a single LCN actuator. We highlight that photochemical actuation can be significantly boosted by photothermal heating, and synergistic use between these two actuation mechanisms can be used to realize reconfigurable shape morphing in the light-actuated LCN structures. We reconfigure a single actuator into six distinct geometries and apply this concept to build a light-fueled gripper which can be programmed to either release an object once irradiation is ceased, or keep it on hold. We anticipate that this facile, easy-to-prepare reconfigurable actuator paves way towards smart micro-robotics with adaptive motion and reprogrammable performance.

## Methods

**Materials.** The LCN actuators are made by photopolymerization of a mixture containing 52 mol% of LC monomer 4-Methoxybenzoic acid 4-(6-acryloyloxyhexyloxy)phenyl ester (Synthon Chemicals), 18 mol% of LC monomer 4[4[6-Acryloxyhex-1-yl)oxyphenyl]carboxybenzonitrile (Synthon Chemicals), 21 mol% of di-acrylate crosslinker 1,4-Bis-[4-(6-acryloyloxyhexyloxy)benzoyloxy]-2-methylbenzene (Synthon Chemicals), 6 mol% of azo crosslinker 4,4'-Bis[9-(acryloyloxy)nonyloxy]azobenzene (Synthon Chemicals), 1.5 mol% of 1,4-Bis(methylamino)anthraquinone (Disperse Blue 14, Sigma Aldrich) and 1.5 mol% of photoinitiator Bis(2,4,6-trimethylbenzoyl)-phenylphosphineoxide (Sigma Aldrich). All molecules were used as received. The monomer mixture was dissolved in dichloromethane and filtered through PTFE syringe filter (Sigma Aldrich, pore size 0.2 µm), after which it was stirred at 80 °C (100 RPM) for 3 h in order to remove the solvent.

**Photopolymerization**. Glass substrates were cleaned by successive sonication in 2-propanol and acetone baths (20 min each), and dried under the flow of nitrogen gas. For planar alignment, two glass slides were spin coated with 1 wt% water solution of polyvinyl alcohol (PVA, Sigma Aldrich; 4000 RPM, 1 min) and rubbed unidirectionally using a satin cloth. After rubbing, the PVA substrates were blown with high-pressure nitrogen to remove dust particles from the surfaces. For splayed alignment, the glass slides were spin coated with PVA as before and a homeotropic alignment layer (JSR OPTMER, 5000 RPM, 1 min), and baked at 180 °C for 20 min. For cell preparation, two coated substrates were fixed together with UV glue (UVS 91, Norland Products Inc., Cranbury, NJ), using spacer particles (Thermo scientific, 10, 20, or 50 µm) to define the cell thickness. The mixture was then infiltrated into the cell on a heating stage at 90 °C and cooled down to 50 °C with a rate of 3 °C min$^{-1}$. An LED (Prior Scientific; 420 nm, 11 mW cm$^{-2}$, 30 min) was used to polymerize the LC mixture. A linear polarizer (Thorlabs, LPVISE100-A, extinction ratio >100) was placed in between the light source and the sample, with orientation perpendicular to the alignment direction in order to minimize the cis-isomer concentration induced by the polymerization. The polymerized film together with cross-polarized microscope images are shown in Supplementary Figure 10. The cell was opened, and strip-like LCNs were cut out from the film using a blade.

**Characterization**. Absorption spectra and isomerization kinetics were measured with a UV–Vis spectrophotometer (Cary 60 UV–Vis, Agilent Technologies) in a 10 µm splay-aligned LCN film. Photoexcitation experiments were conducted using wavelength of 365 nm (Prior Scientific, 18 mW cm$^{-2}$, 1 min) to trigger trans–cis isomerization, and 460 nm (26 mW cm$^{-2}$, 1 min) for cis–trans isomerization. A probe at 385 nm was used to determine the kinetics of cis-isomers. A peltier thermostated cell holder (temperature accuracy ± 0.1 °C) was used to control the sample temperature. Optical images and movies were recorded using a Canon 5D Mark III camera (100 mm lens), and thermal images were recorded with an infrared camera (FLIR T420BX) equipped with a close-up (2×) lens. Cross-polarized microscope images were taken by Zeiss Axio Scope.A1. Differential scanning calorimetry measurement was performed with a Mettler Toledo Star DSC821e instrument, using heating/cooling speeds of 10 °C min$^{-1}$.

**Actuation**. A $4 \times 1 \times 0.02$ mm$^3$ splayed LCN strip was used for testing the photomechanical response upon UV (365 nm, 9 mW cm$^{-2}$), blue (460 nm, 18 mW cm$^{-2}$), and red (0–260 mW cm$^{-2}$) illumination (Prior Scientific). A $25 \times 1 \times 0.05$ mm$^3$ planar-aligned LCN was used for the reconfigurable shape morphing. UV masks were made by hand-cutting black tape and fixing it on a glass substrate. The sample was sandwiched between two masks (or a mask and a substrate) and exposed to UV light (365 nm, 50 mW cm$^{-2}$, 5 s). The shape morphing was induced upon red-light illumination (660 nm, ca. 300 mW cm$^{-2}$ for 0–5 s). For erasing the shape change, the samples were exposed to the combination of blue (ca.100 mW cm$^{-2}$, 460 nm) and red (ca.300 mW cm$^{-2}$, 660 nm) light for 30 s.

**Fabrication of light-fueled gripper**. The gripper was made by fixing two pieces of $5 \times 1 \times 0.05$ mm$^3$ splay-aligned LCN strips onto a tip of a capillary tube using UV glue. Three-dimensional translation stage was used to manually control the position. A laser beam (635 nm, 3 W cm$^{-2}$, RLTMRL-635-1W-5, Roithner Lasertechnik) illuminated the LCN from the back along the capillary direction as shown in Supplementary Fig. 7. To change the gripper property, i.e., to command it to hold the object, the device was irradiated with UV light (365 nm, 50 mW cm$^{-2}$, 10 s).

**Data analysis**. The cis-isomer population was calculated from the absorption spectra shown in Supplementary Fig. 3, according to the Beer–Lambert law ($A = \varepsilon \times b \times c$; where $A$ is absorbance, $\varepsilon$ is molar absorption coefficient, $b$ is thickness, and $c$ is concentration), and assuming 1% cis-fraction at ambient conditions. The lifetime is calculated using a single exponential fitting in the kinetic data. The bending angle is measured by the central angle assuming the bent strip as a perfect arc of a circle.

## Data availability

The data that support the findings of this study are available from the corresponding authors upon request.

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

## Acknowledgements

A.P. gratefully acknowledges the financial support of the European Research Council (Starting Grant project PHOTOTUNE; Agreement No. 679646). H.Z. is thankful to the TUT postdoctoral fellowship program. We are indebted to T. Hukka for assistance with DSC, and Professor Olli Ikkala (Aalto University) for inspiring discussions and insightful comments.

## Author contributions

H.Z. and A.P. conceived the project; M.L. and H.Z. performed experiments. All authors contributed in writing the manuscript.

## Additional information

**Competing interests:** The authors declare no competing interests.

