## [Peer Review File · Nature Communications]

Reviewers' comments:

Reviewer #1 (Remarks to the Author):

This manuscript reports a synergistic use of photochemical and photothermal actuation, which allows the full optical control of shape programming and shape morphing. The photoactive units in the liquid crystal elastomers have distinct wavelength responsiveness which triggers trans-cis isomerization, cis-trans isomerization, and photothermal effect. By using the light with different wavelengths, the actuation of each photoactive effect could be observed, and the combined photochemical and photothermal actuation was quantitatively analyzed. The photochemical deformation was amplified by the photothermal heating, and it is attributed to the photothermal release of photochemically induced stress inside the polymer network. Finally, six different geometries were successfully achieved within the same sample by masked UV exposure, red-light illumination, and blue light exposure. I think it would be suitable for publication in "Nature Communications" after the following points are addressed.

1. Some of the important literatures about reconfigurable actuators are missing; Liu et al, Prog. Polym. Sci. 2016, 52, 79–106; Ko et al, Acc. Chem. Res. 2017, 50, 691–702; Takashima et al, Nat. Commun. 2012, 3, 1270. Authors need to introduce these literatures related to their topics for the readers to understand the research field clearly.
2. Authors utilized strip-like LCN actuators for the fast and reconfigurable actuation in response to the different wavelength of light. However, depending on the dimension of the strip (in particular, thickness), the response time, actuation strength, and multiple responsiveness will be very different. It will be better if the authors can provide additional actuation properties depending on the thickness of LCN strips.
3. This work demonstrates multiple light responsive actuators. The UV, blue, and red lights were utilized for the reversible reconfiguration of actuators. For the practical use, what will be the stability or responsiveness of the suggested actuators under ambient laboratory white light or outside sunlight conditions?
4. In page 5, "The photoactive units were selected based on the following criteria: (i) the wavelengths with which to trigger trans-cis isomerization, cis-trans 100 isomerization, and the photothermal effect must be spectrally separated (see Fig. 2d)" and "The UV-Vis spectra of the polymerized LCN actuator are shown in Fig. 2d, confirming the spectral separation and efficient, reversible isomerization of the azobenzene crosslinks." Please add the detailed explanation of the wavelength of trans-cis isomerization, cis-trans isomerization, and the photothermal effect. In addition, it is hard to understand Fig. 2d based on the above explanations.
5. In figure caption of Figure 2d, please add the detailed information of colored lines about which excitation wavelength was used in which actuation.
6. In page 6, "In splay films, both photochemical and photothermal effects give rise to as schematized in the inset of Fig 3a." However, there is no inset in Figure 3a.
7. In page 6, "Both bending and relaxation take place relatively fast, within 2 s once starting/ceasing the irradiation (see Fig. S4 for the thermal camera characterization)," Please mark the shape of LCN strip in thermal camera images at 0 s to clearly show the original state.
8. In a series of thermal camera images in Figure S4, it seems that the LCN strips are relaxed with the time. From the manuscript, the photothermal heating induces the LCN strips to be bent, however, the behavior in camera images is the opposite.

Reviewer #2 (Remarks to the Author):

The authors present work on the use of photo-chemical and -thermal effects to control actuation of small soft-robots. The science seems to be correct, and the results accurate. However, there is plenty of prior art in the field. Wavelength-selective actuators have been reported with better performance in terms of power consumption and responsivity. Some of the previous work may not

fall in the "soft-robotics" category, but other work from 2014 that uses carbon nanotubes with single chirality is very relevant. Plenty of other relevant work has followed, and the results are constantly improving and advancing the field.

Therefore, this reviewer does not question the validity of the present work, but would like to see the work put into perspective with the already published work, and see the details on the contributions to the state-of-the-art. Nature Comms strives for novelty and significant advances in the field --the authors need to make a compelling argument along those lines.

**Response to reviewer comments for manuscript NCOMMS--18-18727: “Reconfigurable photoactuator through synergistic use of photochemical and photothermal effects”****Reviewer # 1:**

Reviewer: This manuscript reports a synergistic use of photochemical and photothermal actuation, which allows the full optical control of shape programming and shape morphing. The photoactive units in the liquid crystal elastomers have distinct wavelength responsiveness which triggers trans-cis isomerization, cis-trans isomerization, and photothermal effect. By using the light with different wavelengths, the actuation of each photoactive effect could be observed, and the combined photochemical and photothermal actuation was quantitatively analyzed. The photochemical deformation was amplified by the photothermal heating, and it is attributed to the photothermal release of photochemically induced stress inside the polymer network. Finally, six different geometries were successfully achieved within the same sample by masked UV exposure, red-light illumination, and blue light exposure. I think it would be suitable for publication in “Nature Communications” after the following points are addressed.

OUR ANSWER: We would like to thank the reviewer for his/her positive assessment and suggestion for publication. The points raised are addressed case by case below.

Reviewer: 1. Some of the important literatures about reconfigurable actuators are missing; Liu et al, Prog. Polym. Sci. 2016, 52, 79–106; Ko et al, Acc. Chem. Res. 2017, 50, 691–702; Takashima et al, Nat. Commun. 2012, 3, 1270. Authors need to introduce these literatures related to their topics for the readers to understand the research field clearly.

OUR ANSWER: We thank the reviewer for pointing out these important references, and agree that to broaden the perspective of the research background towards other types of stimuli-responsive materials as the ones considered in this work, they should be added to the citation list: they can be found as Refs 14, 15, 16 in the revised version of the manuscript. At the same time, we would like to draw attention to the fact that our reconfigurability scheme is conceptually different from the schemes presented in the literature. To highlight this, we have included a more comprehensive explanation to strengthen the illustration given in Fig. 1, reading now as follows:

“The term reconfigurable actuation is often times used to refer to the ability to reversibly switch between different shapes upon external stimulus^{14,15}. However, conventional stimuli-driven actuators, LCNs among other materials, even if reversible, typically exhibit only one pre-determined state of deformation under one specific stimulus. Hence, to obtain multiple shape changes, the stimulus has to be tuned, which in case of light-triggered actuation can refer to, e.g., changing the light intensity or its spatial/temporal distribution. Obtaining multiple shape changes upon an unchanged stimulus thus requires a programming step during fabrication (such as controlling the photo-polymerization process²⁴), which eventually results in different samples with identical composition but distinct actuation modes, as conceptually outlined in Fig. 1a under “conventional photoactuators”. In contrast, reconfigurable actuators, as we define them, have the ability to obtain multiple shape changes from one single sample upon one identical stimulus, as illustrated in Fig. 1b.”

Reviewer: 2. Authors utilized strip-like LCN actuators for the fast and reconfigurable actuation in response to the different wavelength of light. However, depending on the dimension of the strip (in particular, thickness), the response time, actuation strength, and multiple responsiveness will be very different. It will be better if the authors can provide additional actuation properties depending on the thickness of LCN strips.

OUR ANSWER: We thank the reviewer for an excellent suggestion, motivated by which we repeated the photoactuation experiments similar to those shown in Fig. 3 of the manuscript for samples with thicknesses 20, 30 and 50 μm . The results are given in Fig. S5. The behavior is qualitatively similar for all sample thicknesses, while the differences observed can be attributed to the fact that the photopolymerization process itself depends on the sample thickness. We added following discussions into the revised manuscript to explain the observations shown in Fig. S5:

p. 7: *“Upon photothermal heating (660 nm), the strip exhibits a gradual bending with magnitude increasing as a function of irradiation intensity (red circles in Fig. 3a), accompanied by a linear increase in temperature (blue dots in Fig. 3a). The speed of temperature increase/decrease upon light on/off does not depend on irradiation intensity (Fig. S4), and both bending and unbending take place relatively fast, within less than 2 s once starting/ceasing the irradiation. As shown in Fig. S5 a-c, the response time is similar for samples with different thicknesses.”*

p. 8: *“Such enhanced photoinduced deformations are observed in samples with different thickness, as shown in Fig. S5d, which also illustrates that the bending angle increases with sample thickness, which by hindsight might seem unexpected. We attribute this observation to changes in material rigidity upon changing the thickness, since the polymerization process itself depends on the sample thickness: the thicker the sample, the stronger the absorption due to the azobenzene units at the polymerization wavelength (420 nm; see Fig. 2d for the azobenzene absorption). Thus, the less efficient the polymerization, which eventually leads to enhanced photoinduced bending in the thicker (and softer) sample.”*

Reviewer: 3. This work demonstrates multiple light responsive actuators. The UV, blue, and red lights were utilized for the reversible reconfiguration of actuators. For the practical use, what will be the stability or responsiveness of the suggested actuators under ambient laboratory white light or outside sunlight conditions?

OUR ANSWER: As suggested by the reviewer, we tested the stability of the reconfigured actuators by measuring the thermal relaxation under different environmental conditions: normal laboratory lighting, outdoor sunlight, and 660 nm red-light illumination. While the actuation and relaxation remain constant and repeatable under specified conditions (e.g. temperature, irradiation intensity), the thermal relaxation is faster outdoors than indoors under normal room lighting due to higher rate of *cis-trans* isomerization under sunlight. The result is shown in Fig. S9, and the following text has been added/modified to the discussion part of the manuscript:

“The reconfigurability is based on the spatial control over the concentration of cis-isomers, thus the stability of the reconfigured state being inherently connected to the cis-lifetime of the azobenzene crosslinks. Therefore, the stability of the deformed structure depends on the environmental conditions, being longer in normal room lighting than outdoors under direct

sunlight or at elevated temperatures (Fig. S9). With the present material composition, the bent film recovers 50% of its original shape within 9 min under laboratory lighting (Fig. S9a), revealing the time limit for preserving the reconfigured state.”

Reviewer: 4. In page 5, “The photoactive units were selected based on the following criteria: (i) the wavelengths with which to trigger trans-cis isomerization, cis-trans isomerization, and the photothermal effect must be spectrally separated (see Fig. 2d)” and “The UV-Vis spectra of the polymerized LCN actuator are shown in Fig. 2d, confirming the spectral separation and efficient, reversible isomerization of the azobenzene crosslinks.” Please add the detailed explanation of the wavelength of trans-cis isomerization, cis-trans isomerization, and the photothermal effect. In addition, it is hard to understand Fig. 2d based on the above explanations.

OUR ANSWER: Thanks for the useful suggestion. We have modified Fig. 2d accordingly.

Reviewer: 5. In figure caption of Figure 2d, please add the detailed information of colored lines about which excitation wavelength was used in which actuation.

OUR ANSWER: The following sentence has been added to the caption of Fig. 2d: “The arrows indicate the excitation wavelengths used in the actuation: 365 nm for trans-cis isomerization (purple arrow), 460 nm for cis-trans isomerization (blue arrow), and 660 nm for photothermal heating (red arrow).”

Reviewer: 6. In page 6, “In splay films, both photochemical and photothermal effects give rise to as schematized in the inset of Fig 3a.” However, there is no inset in Figure 3a.

OUR ANSWER: We apologize for the mistake. The inset has now been added.

Reviewer: 7. In page 6, “Both bending and relaxation take place relatively fast, within 2 s once starting/ceasing the irradiation (see Fig. S4 for the thermal camera characterization),” Please mark the shape of LCN strip in thermal camera images at 0 s to clearly show the original state.

OUR ANSWER: Following the reviewer’s suggestion, the dashed lines have been added into Fig. S4 for easy comparison.

Reviewer: 8. In a series of thermal camera images in Figure S4, it seems that the LCN strips are relaxed with the time. From the manuscript, the photothermal heating induces the LCN strips to be bent, however, the behavior in camera images is the opposite.

OUR ANSWER: The films are not relaxed with time, but they bend with time. The misunderstanding is caused by the fact that the strips are initially (i.e. in the dark) somewhat bent due to inner stress generated during photopolymerization, a phenomenon which is expected and has been studied extensively by Broer and coworkers Ref.43. We note that due to anisotropic thermal expansion, this initial bending is to the opposite direction as compared to the photoinduced bending which is the reason why the strip looks to be “relaxed with time” which is not the case. Red arrows have been added to Fig. S4 to clearly point out the direction of photoinduced bending.

**Reviewer # 2:**

Reviewer: The authors present work on the use of photo-chemical and -thermal effects to control actuation of small soft-robots. The science seems to be correct, and the results accurate. However, there is plenty of prior art in the field. Wavelength-selective actuators have been reported with better performance in terms of power consumption and responsivity. Some of the previous work may not fall in the "soft-robotics" category, but other work from 2014 that uses carbon nanotubes with single chirality is very relevant. Plenty of other relevant work has followed, and the results are constantly improving and advancing the field.

OUR ANSWER: We thank the reviewer for his/her positive comments on our manuscript and for the useful suggestion to broaden the introduction. We acknowledge that there is a wealth of publications in the field of light-driven actuators and micro-robotics. Carbon-nanotube-based actuators are among the biggest catalogue of such materials, and continuously draw great attention due to their huge potential in realizing micro-robots. For the sake of comprehensiveness, we have added a brief comparison between LCNs – the topic of the present work – and other materials such carbon-based actuators into the introduction, together with relevant references:

“Several photo-responsive molecules^{11,12} and material systems^{13,14,15} such as hydrogels^{7,16}, carbon-nanotube-based actuators¹⁷, and shape memory polymers¹⁸ have been explored for devising soft actuators, yet photoresponsive liquid crystal elastomers and polymer networks (referred to as LCNs from hereon) are in many cases particularly intriguing¹⁹. LCNs combine the anisotropy of liquid crystals and the elasticity of loosely crosslinked polymeric materials, and by “programming” the molecular alignment pattern during the fabrication process (photopolymerization), an overwhelming variety of light-induced shape changes (e.g. contraction, bending, coiling, buckling, etc.) can be obtained^{20,21,22,23}. In contrast to hybrid actuators relying on, e.g., bi-layer structures to obtain efficient actuation^{7,17}, LCNs allow complex 3-dimensional deformation in a monolithic film with uniformly distributed chemical composition¹⁹.”

We also added the following outlook paragraph into the discussion, in order to emphasise the different classes of light-responsive materials and the potential synergies that could be achieved by combinations therein:

“We also emphasize that apart from LCNs, various types of materials exhibit light-induced shape changes, such as hydrogels^{7,16}, temperature-responsive and shape memory polymers^{18,53} functional carbon-nanotube-based bilayers¹⁷, photochromic crystals⁵⁴ etc. While chemists and material scientists continuously extend the diversity of materials and shape changes, engineers and micro-roboticists are dedicated to issues such as energy consumption⁵⁵ and integration of the photomechanical units as part of active circuits². We note that the physical principles underlying light-driven device/robot realization using different materials are rather similar, as exemplified by, e.g., self-shadowing-induced oscillation^{17,56}, snapping motion^{57,58} and rolling locomotion^{17,59}, all achieved in both LCNs and bilayer actuators. Keeping this in mind, we believe that the reconfigurability concept proposed herein can be extended also beyond LCNs, and synergies can be sought for by combining different types of light-responsive materials, to simultaneously harness the strengths of each.”

Finally, we would like to emphasise that the ultimate goal of this manuscript is – for the first time to the best of our knowledge – to present a reconfiguration scheme (as defined in Fig. 1b) for photoactuation by purely optical means, through synergistically using photochemical and photothermal effects in LCNs. The proposed scheme is conceptually different from any other that has appeared in the literature, aiming towards multiple shape changes within one single monolithic structure under an identical stimulus. Reconfigurability is very significant for adaptive microrobotics, yet very rarely achieved in any types of, and, to our best knowledge, not yet demonstrated in carbon-based actuators, which however have other advantages as compared to LCNs. We believe that synergistically harnessing the strengths of different types of light-driven actuators by combining several materials is one of the important goals of light-driven soft robotics of tomorrow.

REVIEWERS' COMMENTS:

Reviewer #1 (Remarks to the Author):

Authors addressed all the comments raised by the reviewers. It is now appropriate for publication.

Reviewer #2 (Remarks to the Author):

The authors have made genuine and significant effort to address all the points raised in the first version. Thus, I would like to start by praising the authors in this task. In particular, the addition of Fig.1 is probably the most important addition to the general reader. That figure is crucial and draws a very clear map of that summarizes the manuscript.

However, just with a quick search, I found a very relevant article (DOI: 10.1126/sciadv.1602697) which was not mentioned. This article includes programming of multiple states in micro actuators by using different wavelengths. They also provide a study on the effects of film thickness, as raised by the first reviewer as well. Thickness could end up masking effects caused by radiation wavelength. They also provide a study on power consumption (which by the way is a topic that was not commented in either version of the manuscript). The authors should briefly report and compare these. Even if the numbers in terms of power consumption and efficiency are not better, that will not affect the novelty and contribution of the article --there is plenty of high quality science in the manuscript for Nature Comms. However, in the interest of being comprehensive and provide the most complete and broaden the scientific contribution of the manuscript, I would strongly advise the authors to include this short and minor change.

Response to reviewer comments for manuscript NCOMMS--18-18727A:

“Reconfigurable photoactuator through synergistic use of photochemical and photothermal effects”

Reviewer #1:

Authors addressed all the comments raised by the reviewers. It is now appropriate for publication.

Reviewer # 2:

Reviewer: The authors have made genuine and significant effort to address all the points raised in the first version. Thus, I would like to start by praising the authors in this task. In particular, the addition of Fig.1 is probably the most important addition to the general reader. That figure is crucial and draws a very clear map of that summarizes the manuscript.

However, just with a quick search, I found a very relevant article (DOI: 10.1126/sciadv.1602697) which was not mentioned. This article includes programming of multiple states in micro actuators by using different wavelengths. They also provide a study on the effects of film thickness, as raised by the first reviewer as well. Thickness could end up masking effects caused by radiation wavelength. They also provide a study on power consumption (which by the way is a topic that was not commented in either version of the manuscript). The authors should briefly report and compare these. Even if the numbers in terms of power consumption and efficiency are not better, that will not affect the novelty and contribution of the article --there is plenty of high quality science in the manuscript for Nature Comms. However, in the interest of being comprehensive and provide the most complete and broaden the scientific contribution of the manuscript, I would strongly advise the authors to include this short and minor change.

OUR ANSWER: We would like to thank the reviewer for his/her appreciation about our effort in revision, as well as for pointing this important reference. In the revised version, we have mentioned the thickness-dependent photoactuation in carbon-nanotube-based actuator, as a comparison with our liquid-crystal-based actuator. It now reads:

p. 8: *As a comparison, in carbon-nanotube-based photothermal actuators both the degree of deformation and wavelength sensitivity decrease when increasing the thickness of the actuator⁴⁴. Herein, we attribute the slightly enhanced actuation in thicker LCN strips Supplementary Figure 5d to changes in material rigidity upon changing the thickness.*

AND

p. 11: *The demonstrated reconfigurable photoactuator is unique compared to other dual-responsive liquid crystalline polymer systems reported in the literature^{46,47,48}, and to wavelength-selective carbon-nanotube-based photothermal actuators^{17,44}.*

Regarding to the power consumption, we would like draw the attention to the fact that comparison between different actuating systems is very tricky, because the operation environment and dimensions of the structure strongly affect the photothermal process, often leading to a big variation of the absolute value of power consumption. The typical LCN sample used in this study is operated under laser illumination with power around 300 mW (spot size 1 cm², thus 300 mW cm⁻²) at room temperature, thus total power consumption is 300 mW. The power consumption in the paper mentioned by the referee is one order of

magnitude lower, 30-60 mW in their miniature structure, which is ten times smaller than our case (a smaller structure requires less power). We would like to note that in the reference pointed out the authors first raised up the sample temperature to 40 °C, close to the phase transition temperature. Second, they illuminated the sample with a focused laser beam (spot diameter is 500 μm , page 3 in SI), resulting in a light intensity above 15 W cm^{-2} that actually is 3 orders of magnitude higher than in our case. In other words, if the power consumption per unit area is calculated, the mm-sized LCN actuator consumes 0.3 W cm^{-2} , which is significantly less than the micrometre-sized actuators triggered with focused laser beams. However, we would prefer avoiding the comparison of the values of power consumption between different systems and different experiments. Therefore, we are inclined to not discussing this issue in the manuscript. In the manuscript, each experiment is detailed with the data of excitation light intensity and the sample size, which defines the light-absorbing surface. Thus, we believe the total power consumption in each experimental demonstration is clear to the readers.